# The Metabolism of Reactive Oxygen Species and Their Effects on Lipid Biosynthesis of Microalgae

**DOI:** 10.3390/ijms241311041

**Published:** 2023-07-03

**Authors:** Liufu Wang, Tian Yang, Yingying Pan, Liqiu Shi, Yaqi Jin, Xuxiong Huang

**Affiliations:** 1Centre for Research on Environmental Ecology and Fish Nutrition (CREEFN) of the Ministry of Agriculture and Rural Affairs, Shanghai Ocean University, Shanghai 201306, China; m160674@yzu.edu.cn (L.W.);; 2College of Animal Science and Technology, Yangzhou University, Yangzhou 225009, China; 3Building of China—ASEAN Belt and Road Joint Laboratory on Mariculture Technology and Joint Research on Mariculture Technology, Shanghai 201306, China; 4National Demonstration Center for Experimental Fisheries Science Education, Shanghai Ocean University, Shanghai 201306, China

**Keywords:** oxidative stress, regulatory mechanisms, lipid accumulation, oleaginous microalgae

## Abstract

Microalgae have outstanding abilities to transform carbon dioxide (CO_2_) into useful lipids, which makes them extremely promising as renewable sources for manufacturing beneficial compounds. However, during this process, reactive oxygen species (ROS) can be inevitably formed via electron transfers in basal metabolisms. While the excessive accumulation of ROS can have negative effects, it has been supported that proper accumulation of ROS is essential to these organisms. Recent studies have shown that ROS increases are closely related to total lipid in microalgae under stress conditions. However, the exact mechanism behind this phenomenon remains largely unknown. Therefore, this paper aims to introduce the production and elimination of ROS in microalgae. The roles of ROS in three different signaling pathways for lipid biosynthesis are then reviewed: receptor proteins and phosphatases, as well as redox-sensitive transcription factors. Moreover, the strategies and applications of ROS-induced lipid biosynthesis in microalgae are summarized. Finally, future perspectives in this emerging field are also mentioned, appealing to more researchers to further explore the relative mechanisms. This may contribute to improving lipid accumulation in microalgae.

## 1. Introduction

Due to large emissions and climate hazards, how to reduce carbon dioxide (CO_2_) is of great concern to all societies. Microalgae strains can be good tools for reducing CO_2_, as they have excellent adaptation to severe environments and high-concentration C sources as well as fast-growing ability while fixing CO_2_ [1]. In addition, microalgae can convert CO_2_ into biological compounds with high value (e.g., lipid or protein, etc.). Therefore, while using photoautotrophic microalgae to reduce CO_2_, increasing the byproduct yield was considered a win–win method to improve the sustainable development of the microalgae–CO_2_ fixation industry [2,3,4].

Among these products of microalgae, lipids can accumulate up to 30–70% of content in cell dry weight [5]. Microalgal lipids mostly consist of saturated and monounsaturated fatty acids, which are hopeful replaceable sources for biodiesel manufacture [6]. Meanwhile, a certain species of microalgae produce polyunsaturated fatty acids (PUFAs) including eicosapentaenoic acid (EPA) and docosahexaenoic acid (DHA). Recently, their positive impacts on the health of humans and animals have made these PUFAs a subject of intense interest. However, regardless of the application purposes, the commercialization of lipid production in microalgae faces various challenges due to low productivity. To alleviate the problem, many studies have focused on increasing lipid content in microalgae [7,8].

Reactive oxygen species (ROS) mainly include singlet oxygen (^1^O_2_), hydroxyl radical (·OH), superoxide (O_2_^−^), and hydrogen peroxide (H_2_O_2_). ROS are primarily produced in the location of metabolisms featured as high redox potentials (e.g., electron transport chains (ETCs) of chloroplasts/mitochondria) [9,10]. Specifically, ROS are derivations of ETCs, which come from the combination of electron leakages and molecular oxygen (O_2_). When microalgae are subjected to environmental stresses, these ETCs may be inhibited, leading to more electron leakages and ROS formation. Although these ROS were initially thought to be harmful coproducts that had to be erased by scavenging systems, more recent research has demonstrated that ROS at low concentrations are utilized as important signaling molecules in most organisms. It was also revealed that a basal content of ROS is required for life, which serves as a set of safe signal molecules that regulate many different biological processes, such as the increase in lipid accumulation in microalgae [11].

ROS can oxidize proteins via the oxidation of cysteine (Cys) or methionine (Met) residues. In these ways, they are able to further regulate various proteins as well as affect different protein phosphorylation relays and transcription factors (TFs), etc. [11]. The changes induced by ROS may be connected to the lipid increases in microalgae (Table 1) [12,13,14,15,16,17,18,19]. In this review, the generation and scavenging systems of ROS in microalgae (Section 2 and Section 3) as well as lipid biosynthesis pathways (Section 4) are illustrated with a discussion on how ROS affect signaling pathways in lipid biosynthesis (Section 5). In addition, the strategies and applications of ROS-induced lipid biosynthesis (Section 6) as well as future perspectives in this field (Section 7) are mentioned. These may be conducive to making us more familiar with the roles of ROS on the lipid production of microalgae, so as to promote further research in this field [20,21].

## 2. ROS Production in Main ETCs of Microalgae

Four electrons are required for reducing O_2_ to a water molecule (H_2_O) completely. O_2_ tends to a gradually univalent route of reduction, which leads to partially reduced intermediates (Figure 1). OH can be produced by the interaction of O_2_^−^ and H_2_O_2_ via metal ions. O_2_^−^ is the conjugate base of the perhydroxyl radical (HO_2_). Therefore, under acidic situations, the highly reactive HO_2_ may be predominant, while the O_2_^−^ predominates under alkaline conditions. About 22 kcal/mol energy is used to make O_2_ from the ground state to the first singlet state [22]. The general chemical processes of various ROS production, and the main ETCs structure and related ROS production in microalgae are summarized below.

### 2.1. Introduction to the Main ETCs

*Cyanobacteria* are a group of prokaryotic microalgae that are able to conduct oxygenic photosynthesis and respiration at the same time and in the same compartment [23]. The thylakoid membrane, an internal system present in nearly all *Cyanobacteria*, not only separates the cytoplasm from the lumen, but also has two types of ETCs for photosynthesis and respiration. The two ETCs intersect, and the same components in the membrane are partly utilized by them (Figure 2). Photosystem II (PS II) and photosystem I (PS I) are specific complexes that participate in photosynthetic electron transfer, whereas the type 1 NADPH dehydrogenase (NDH-1), succinate dehydrogenase (SDH), and terminal oxidase (Ox) are specific for respiratory electron flow. Those shared by both pathways are plastoquinone (PQ), cytochrome b6f complex (cyt b6f) and plastocyanin (PC). Note that the cytoplasmic membrane of most *Cyanobacteria* only has respiratory ETCs without photosynthetic ETCs. In consequence, photosynthetic electron transport appears only in the thylakoids of most *Cyanobacteria*, while respiratory electron transport appears in the cytoplasmic membrane system as well as in the thylakoid system [24].

Eukaryotes microalgae are different from *Cyanobacteria* in that photosynthetic and respiratory bioenergetic pathways are spatially separated [25]. Chloroplasts are made up of envelope, stroma and thylakoid. The thylakoid membrane, also known as the photosynthetic membrane, has a photosynthetic ETC. Reaction centers PSII and PSI are interactive via cyt b6f and two mobile electron carriers (i.e., PQ and PC) (Figure 3). In eukaryotes microalgae, the energy of light quanta caught by PSII and PSI is shifted into the energy of separated charges, which catalyzes electron transfers from the H_2_O to oxidized form of nicotinamide adenine dinucleotide phosphate (NADP^+^): H_2_O→PSII→PQ→b6f→PC→PSI→NADP^+^. In addition, mitochondria have an inner membrane and outer membrane. The outer membrane is smooth and serves as the boundary of organelles. The inner membrane is folded inward for forming mitochondrial cristae, which has respiratory ETCs (Figure 4). There are five complexes, I–V, in respiratory ETC, which are the reduced form of nicotinamide adenine dinucleotide quinone oxidoreductase (i.e., NADH-quinone oxidoreductase), succinate dehydrogenase, coenzyme Q: cytochrome C oxidoreductase, cytochrome oxidase and adenosine triphosphate synthase (i.e., ATP synthase). These complexes are responsible for electron transport from NADH to O_2_, which is accompanied by oxidative phosphorylation and ATP production [26].

### 2.2. ROS Production in the Main ETCs

Just as with all aerobically living organisms, O_2_ spreads into the cell, which is reduced to ROS (i.e., O_2_^−^ and H_2_O_2_) by the flavoproteins in the respiratory ETC of *Cyanobacteria* [12,27]. In addition, *Cyanobacteria* also face ROS generated by the photosynthetic ETC. Due to the reception of energy from photosensitized chlorophyll, O_2_ becomes ^1^O_2_, which is thought to prevent the repair of inactivated PSII, which reduces the electron transport efficiency. When the electrons in transport chains cannot be fully utilized, they leak out and form O_2_^−^ with O_2_ in photosystem I (PSI) [28,29]. The O_2_^−^ is disproportionated to H_2_O_2_ and O_2_. Then, H_2_O_2_ is reduced to either H_2_O or ·OH [20].

The primary sources of ROS within eukaryotes microalgae are chloroplasts and mitochondria. Chloroplasts are considered to derive from *Cyanobacteria* and their generation mechanisms of ROS from ETCs are similar [30]. Therefore, the ROS sources of chloroplasts will not be repeated, but the relevant mechanism of mitochondria is explained in detail below. In the process of cellular respiration, O_2_ reacts with the electrons from the ETC and produces O_2_^−^ [31]. NADH-quinone oxidoreductase and coenzyme Q: cytochrome C oxidoreductase are the main sites where electrons are transferred to O_2._ The O_2_^−^ is formed by electrons and O_2_ is released into the intermembrane space or matrix and participates in redox reactions [32]. As a charged species, O_2_^−^cannot diffuse across the mitochondrial membrane. Thus, it seems that the voltage-dependent mitochondrial anion channels can facilitate the release of intermembrane mitochondrial O_2_^−^ to the cytosol. In addition, the released O_2_^−^ is rapidly shifted to H_2_O_2_ [33].

## 3. ROS Elimination Systems

Microalgae have complete scavenging systems, which importantly work in decreasing ROS levels to maintain their homeostasis (Figure 5). Some of them are enzymes such as superoxide dismutase (SOD), catalase (CAT), ascorbate peroxidase (APX), glutathione peroxidase (GPX), peroxiredoxins (PrXs) and so on. Others are nonenzymatic antioxidants, including carotenoid, glutathione (GSH), ascorbic acid (AsA), etc.

### 3.1. Enzyme Scavenging System

SOD, as the first defensive line, are vulnerable to ROS. They are major intracellular antioxidant metal enzymes and mainly divided into three categories: copper/zinc SOD (Cu/Zn-SOD), iron SOD (Fe-SOD) and manganese SOD (Mn-SOD). They have different protein structures, which catalyze the conversion of O_2_^−^ to H_2_O_2_ through disproportionation in various organelles [34].

CAT contains four subunits, which have heme iron groups bound to their active sites. CAT, whose cofactors are iron or manganese, has high catalytic efficiency and can decompose millions of H_2_O_2_ to H_2_O and O_2_ per 1 s in peroxisome [35].

APX is the only enzyme to remove H_2_O_2_ in chloroplasts and cytoplasm through participation in the water–water cycle and ascorbic acid–glutathione (AsA–GSH) cycle [36]. It is able to use AsA as the hydrogen donor to decompose H_2_O_2_ into H_2_O and monodehydroascorbate and can also affect quantum efficiency to control the associated electron transport in the AsA–GSH cycle for the regulation of H_2_O_2_ decomposition.

GPX has a highly reactive thiol group that can catalyze H_2_O_2_ to H_2_O via glutathione (GSH) and glutathione disulfide (GSSG) [37]. For example, under nitrogen starvation for 24 h, the ROS content in the GPX5-deficient strain of *Chlamydomonas reinhardtii* increased by 1.5 times that in the parent algal strain CC4348 [38].

PrXs, another family in peroxidases, can reduce various peroxides such as alkyl hydroperoxides and peroxynitrite [39]. PrXs, different from SOD and CAT, have no specific cofactors. They rely on the thiol–disulfide transition in Cys controlled by electron donors (e.g., thioredoxin and cyclophilin) to convert H_2_O_2_ into H_2_O [40]. Since their high vitality and capacity to make use of extensive substrates, PrXs are thought as the predominant peroxidases in most living organisms [41].

### 3.2. Nonenzymatic Antioxidants

Carotenoids protect the light-harvesting complex and maintain the stability of thylakoid membranes in four main ways: (i) removing ^1^O_2_ and dissipating energy as heat; (ii) reacting with ^3^Chl* or excited state chlorophyll (Chl*) molecules to prevent the formation of ^1^O_2_; (iii) consuming excess excitation energy through the lutein cycle; (iv) creating a layer of lipid droplets for photoprotection [42].

GSH is a thiol tripeptide composed of glutamic acid, cysteine and glycine. It is widely present in plant organelles such as chloroplasts, mitochondria and endoplasmic reticulum. It mainly uses the AsA–GSH cycle to clear ROS and maintain cellular redox homeostasis [35,36]. The AsA–GSH cycle is a major antioxidant defense way of plant cells, where AsA and GSH as electron donor is the most abundant nonenzymatic antioxidant. AsA can not only remove ROS through the ASA–GSH cycle but also protect cells from oxidative damage through α-tocopherol regeneration [36].

Besides antioxidants, other compounds (e.g., proline, flavonoids and fatty acids) with low molecular weights may also have significant antioxidant abilities [43]. For instance, when microalgae were under the conditions of nutrient (P and N) replete, the content of phenolic compounds decreased obviously but the concentrations of ASA and tocopherol were higher than those in normal conditions [44]. It was also reported that phenolic compounds are contributors to antioxidant activities of several microalgae species such as *Phaeodactylum* sp., *Chaetoceros* sp., *Skeletonema* sp., *Isochrysis* sp. and *Odontella* sp. [45]. In particular, in a certain species in the genera of *Desmodesmus*, *Phaeodactylum*, *Dunaliella*, *Chlorella* and *Nannochloropsis*, these compounds play major roles in the total antioxidant activity [46]. Moreover, tocopherol can work as an antioxidant by two reactions: ROS oxidize tocopherol to a tocopheryl radical; meanwhile, ^1^O_2_ is converted to H_2_O_2_ [20]. It is also known that tocopherol participates in avoiding membrane lipid peroxidation [47].

## 4. Key Pathways of Lipid Biosynthesis in Microalgae

Although there are many kinds of lipids in microalgae, only key fatty acids and triacylglycerol (TAG) seem to have application potentials. Therefore, it is necessary to understand their processes of biosynthesis [48,49]. To be specific, the carbon fixed by photosynthesis in microalgae is first used in chloroplasts to synthesize fatty acids and then to produce fatty acyl-coenzyme A (acyl-CoA). The acyl-CoA enters endoplasmic reticulum (ER), which reacts with glycerol 3-phosphate (G3P) to form TAG. The general processes are shown in the following paragraphs.

### 4.1. Fatty Acid Biosynthesis in Chloroplasts of Microalgae

In chloroplasts of microalgae (Figure 6), the CO_2_ absorbed is fixed in 3-phosphoglycerate (3-PGA) using Rubisco ribulose-1,5-bisphosphate carboxylase (Rubisco). 3-PGA is transferred to 2-phosphoglyceric acid (2,3-DPG), which is used to produce pyruvate. The pyruvate is shifted to acetyl-coenzyme A (acetyl-CoA), which is catalyzed using pyruvate dehydrogenase (PDHC). Acetyl-CoA is then affected by carboxylase to form malonyl-coenzyme A (malonyl-CoA). This is the first step in fatty acid biosynthesis and requires the participation of ATP and HCO_3_^−^ [50]. Then, malonyl is transferred from coenzyme A to acyl carrier protein (ACP), which results in the formation of malonyl-ACP. Malonyl-ACP, catalyzed by fatty acid synthase (FAS), is used to form fatty acids through a series of reactions of carbon chain lengthening and desaturation. These obtained fatty acids are precursors to TAG biosynthesis, which are transformed to acyl-CoA via long-chain acyl-CoA synthetase (LACS) [51].

### 4.2. TAG Biosynthesis in ER of Microalgae

Acyl-CoA passes through chloroplasts and enters ER for TAG biosynthesis (Figure 7). Moreover, dihydroxyacetone phosphate (DHAP) is first transformed to G3P through dehydrogenase. Then, acyl-CoA and G3P react with acyltransferase to form lysophosphatidic acid (LPA). LPA is combined with acyl-CoA again to produce phosphatidic acid (PA) through acyltransferase. Finally, PA is transformed to TAG via phosphatidic acid phosphatase (PLA) and diacylglycerol acyltransferase [52].

## 5. The Roles of ROS in Microalgal Lipid Biosynthesis Signaling Pathway

### 5.1. ROS Sensing by Receptor Proteins

All living systems catch and cope with information through various receptors located in plasma membranes at the cellular level [53,54,55,56]. In plants, two different transmembrane kinases are found to sense ROS, including histidine kinases (HiKs) and receptor-like protein kinases (RLKs) [57,58,59,60]. The HiKs and response regulator protein (RR) are conserved kinase proteins, which constitute a ‘two-component’ system. When HiKs senses ROS, it may accomplish autophosphorylation at histidine (His) residue. Then, the phosphotransfer from HiKs to RR (aspartate, Asp residues) gives rise to the activation of RR and the output response of the signaling pathway [61]. Moreover, a representative RLK protein includes the protein kinase catalytic domain (PKC) and transmembrane domain (TM) as well as extracellular ligand-binding domain (ECLB). PKC undertakes to send the signals downstream. TM is responsible for connecting ECLB and PKC while ECLB is the central place of the signal combination [62]. ROS may play key roles in the activation of RLKs. ROS molecules, released during stress conditions, can do harm to cell wall components, whose breakdown products may work as ligands for RLKs [63]. In addition, ROS molecules may activate RLKs directly via redox modifications of Cys. Finally, the receipt of ROS by HiKs or RLKs may lead to the production of calcium ion (Ca^2+^) signals and the stimulation of a phospholipase C/D (PLC/PLD) vitality, which triggers the generation of phosphatidic acid (PA). Then, PA and Ca^2+^ are considered to be capable of activating the protein kinase OXL1, the result of which contributes to the stimulation of a mitogen-activated protein kinase (MAPK) cascade [64,65,66].

From unicellular organisms to complex organisms, the MAPK function and regulation of the cascade exists universally [67]. The MAPK signaling cascade is generally made up of three enzymes: MAPK kinase kinase (MEKK), MAPK kinase (MEK), and MAPK. As soon as responding to appropriate signals, the MEKK–MEK–MAPK cascade is sequentially activated through phosphorylation. Then, MAPK-stimulated phosphorylation of substrate proteins serves as a valve to turn on or off the vitality of the TFs, which regulate the gene expression of lipid biosynthesis [68].

Many studies show that various stresses trigger ROS accumulation that in turn activates MAPK cascade regulation, which is reported to be a center of regulating cellular answers to multiple stresses in plants [69]. Choi et al. [70] was also the first to report ROS (H_2_O_2_)–MAPK pathways involved in the metabolism of lipid in microalgae. Then, Tengsheng et al. [12] evaluated the influences of the combination of sodium chloride (NaCl) stress and H_2_O_2_ on lipid biosynthesis in *Monoraphidium* sp. QLY-1. Their results offered evidence that under the condition of abiotic stress, crosstalk between ROS and Ca^2+^ can activate MAPK signaling cascades to regulate lipid biosynthesis processes in microalga [71].

### 5.2. ROS Inhibition of PPs

Protein phosphorylation, as a key reversible posttranslational modification that participates in regulating many intracellular processes, takes place in a coordinated manner through two types of enzymes (i.e., kinases and PPs). Specifically, kinases shift the γ-phosphoryl group of ATP to proteins, whereas PPs dephosphorylate the phosphoproteins [72]. To provide a typical example, the vitality of MAPK shows a balance between the activities of the kinases (reviewed above) and PPs. Since both phosphorylation of threonine and tyrosine residues is necessary for the activation of MAPK, dephosphorylation of either can result in the inactivation of MAPK. This can be realized by tyrosine-specific PPs (PTPs), serine/threonine-specific PPs (PP2Cs) or by dual specificity PPs (DSPs and their subclass, MKPs) [73].

Several reports have shed light on the fact that the invertible oxidation of PPs through reactive Cys residues alters their function and downstream signaling components that then trigger intracellular responses. Specifically, these PPs may possess particular domains that sustain the reactivity of Cys, allowing for interactions with ROS. The Cys contains a thiol moiety (-SH) in its side chain, which is prone to oxidation and facilitates the formation of disulfide (S-S) bonds by combing with another -SH. The S-S bonds produced can then be reversibly reduced to free -SH in cells [74]. Although such reactions are thought to be nonspecific, recent studies have revealed the selective oxidation of Cys residues by ROS in nature. For example, it was found that ROS (H_2_O_2_) can lead to the reversible inhibition of these PPs, thereby shifting between active and inactive states. The PPs can determine the magnitude of reactions in MAPK signaling pathways, thereby regulating adaptive stress responses as well as lipid biosynthesis processes [75].

### 5.3. ROS Activation of TFs

The changes in lipid biosynthesis that arise from fluctuations in intracellular redox conditions appear to be regulated by various TFs. For instance, Hu et al. [76] found that there are 11 lipid-related TF families predicted in microalgae *Nannochloropsis* IMET1, where the basic leucine zipper (bZIP) and v-myb avian myeloblastosis viral oncogene homolog (MYB) families were also activated in oxidative stress resistance. It is also shown that Atf1 TF in the bZIP family is controlled by an MAPK cascade and takes part in the modulation of secondary metabolism [77]. Under the conditions of regular growth, N limitation and high sea salt, NsbZIP1, as another TF in bZIP family was overexpressed; then, the neutral lipid contents were increased by 33%, 203% and 88% in *Nannochloropsis salina*, respectively [78]. Furthermore, other TFs, such as AP1 and CRC, are motivated by ROS directly and regulate respective genes in the biosynthesis of fatty acid, such as *BCCP2* gene encoding an ACCase, *At3955310* gene encoding ketoacyl-ACP reductase and other FA guide genes [79]. Furthermore, TFs in the families of MYB and bHLH are also reported to be modulated by ROS and related to lipid biosynthesis [76,80].

Compared to other ROS, H_2_O_2_ is more steady and is able to pass through membranes via aquaporins, which are advantages in terms of its signaling ability. Therefore, the impacts of H_2_O_2_ on TFs are relatively well explored. For instance, some studies have reported that the bZIP family can be activated by H_2_O_2_ through modifying cysteine thiol groups directly [81]. The thioredoxin system is also reported to participate in the redox regulation of AP1 transcriptional activity. As we all know, the activation of TFs might result in its interaction with cis-regulatory elements to modulate cellular metabolisms and signaling pathways [82]. These studies show that the transduction of H_2_O_2_-based signals depends on sulfur chemistry, with the main player being the reversible oxidation of sulfur-containing groups (e.g., Cys residues and thioredoxin) in TFs, which in turn may affect lipid biosynthesis in microalgae [83].

## 6. Strategies and Applications of ROS-Induced Lipid Biosynthesis in Microalgae

Lipid biosynthesis in microalgae induced by stress may be accompanied by altered levels of antioxidant substances or intracellular ROS. When *Acutodesmus dimorphus* was continuously cultured at 25, 35 and 38 °C, the contents of H_2_O_2_ and malondialdehyde in 25 and 38 °C groups were significantly higher than those in the control group (35 °C group). Furthermore, the activities of CAT and APX in the 25 °C group were 1.35 and 1.42 times those in the control group, respectively. CAT and APX activities in the 38 °C group were 2.60 and 1.71 times higher than those in the control group [84]. In recent years, various new detection tools and techniques have realized the application of appropriate exogenous ROS to trigger microalgal lipid accumulation. For example, specific imaging tools (e.g., small molecule fluorescent probes and gene-encoded redox probes) can characterize subcellular localization and ROS fluxes [12,85]. Advanced omics techniques, such as proteomics, genomics, transcriptomics, lipidomics and metabolomics, have also paved the way for the study of cellular oxygenated reductive metabolism and its close links to biological structure and function [86]. A genetically encoded fluorescence sensor can monitor H_2_O_2_ in all major subcompartments of *C. reinhardtii* [87].

At present, the strategies involving ROS to induce lipid biosynthesis in microalgae mainly focus on three aspects: (i) the use of H_2_O_2_, ^1^O_2_ and some nanomaterials to directly trigger lipid accumulation in microalgae; (ii) other stress conditions inducing the increase of ROS content in microalgae cells, which indirectly promote microalgae lipid biosynthesis; (iii) the synergistic effect of the above two. For example, *Chlorella vulgaris* was treated with H_2_O_2_ concentrations of 2 and 4 mmol·L^−1^, and the lipid content was 20% and 87% higher than that of the control group (without H_2_O_2_) after 8 days, respectively [88]. When treated with NaCl (171.12 mmol·L^−1^) or H_2_O_2_ (1 mmol·L^−1^ and 2 mmol·L^−1^), the lipid contents of *Monoraphidium* sp. QLY-1 in coordinated treatment groups (NaCl + H_2_O_2_) were 18.72% and 24.09% higher than that in the single salt stress group, respectively [12]. *Scenedesmus* sp. M22 with high lipid yield and high biomass was screened using ultraviolet mutagenesis. After treatments of 2 mmol·L^−1^ H_2_O_2_, the lipid content in the mutant strain was thrice that of the wild strain [17].

## 7. Future Perspectives

Microalgae may have complete and complex networks for tuning lipid biosynthesis signaling through ROS. However, interesting areas of oxidative stress-induced signal pathways in microalgae are just springing up. Fortunately, microalgae (e.g., *C. vulgaris*, *Synechocystis* sp. PCC6803), as a unique model, can be used to better understand the ROS–lipid mechanism in the future [89,90]. Therefore, we highlight what we believe are the major challenges in this area for future research.

Although the main mechanisms of ROS production and elimination have been elucidated in this paper, it remains unclear how these mechanisms change and further alter ROS homeostasis under stress conditions. Moreover, most previous studies of signal pathways have focused on H_2_O_2_, but little attention has been paid to other ROS. However, recent studies have shown that other ROS (such as ·OH) also play nonnegligible roles in the signaling pathway of microalgal lipid biosynthesis [19]. So, we should devote ourselves to clarifying each ROS metabolism and its biological influences, specific targets as well as subsequent responses [91].

The extranuclear receptors and ROS recognition are inextricably linked. However, in contrast to systematic studies in higher plants, it seems that studies of the relationship between ROS and receptors in microalgae have only just commenced. Studies of ROS and PPs in microalgae are also scarce. In the future, it will be of great significance to find more extranuclear sensors of ROS in microalgae [66]. We expect that advances in ROS-sensitive probes that enable high-resolution visualization of ROS may promote these studies. In particular, ROS-dependent responses are intertwined with intracellular signaling controlled by the RLK. Therefore, it is of importance to identify components that work independently of ROS production. Future explorations of the interplay between these responses can lead to a better understanding of ROS signaling specificity in microalgae [92].

Compared with higher plants, studies on lipid biosynthesis pathways in microalgae are very few. Most knowledge on fatty acids and TAG biosynthesis in microalgae shown above results from studies of higher plants. Previous studies showed that many genes of lipid biosynthesis in higher plants are homologous to those in microalgae. Meanwhile, many enzymes isolated from microalgae have similar biochemical characteristics to those in higher plants. Therefore, it is believed that the pathways of microalgal fatty acids and TAG biosynthesis are basically the same as those in higher plants [93]. However, individual cells of microalgae can perform all tasks from CO_2_ fixation to fatty acids and TAG biosynthesis, while the biosynthesis of fatty acids and TAG in higher plants occurs only in specific tissues or organs (such as seeds or fruits). In addition, microalgae can also produce long-chain polyunsaturated fatty acids (such as EPA and DHA) that are not found in ordinary higher plants. These phenomena suggest that lipid biosynthesis of microalgae may also be slightly different from that of higher plants, which needs additional research to be explored [94].

Recently, a growing number of researchers have realized that the modulation of metabolic pathways should be explored in the background of the entire cell, rather than at the level of individual pathways. Although we have preliminarily summarized the role of ROS in regulating lipid biosynthesis in microalgae (Figure 8), these findings are far from enough to support the rapid development of the microalgae lipid industry. In particular, the application of TFs to control multiple enzymes for the formation of target products has triggered extensive interest [95]. This technology is called transcription factor engineering and can employ the overexpression of TFs that up- or downregulate the pathway(s) involved in the production of lipids. Although transcription factor engineering is beneficial, its current developments are restricted due to a lack of thorough endogenous TF networks [96]. Therefore, it is particularly important to find more TFs and study their mechanism with ROS.

Although the positive aspects of ROS in lipid accumulation are highlighted above, its negative effects should not be ignored either. The lipid accumulation in *Monoraphidium* sp. QLY-1 was shown to be inhibited by the combined treatment of 171.12 mmol·L^−1^ NaCl and 4 mmol·L^−1^ H_2_O_2_ [12]. *C. vulgaris* was treated with exogenous addition of 6 mmol·L^−1^ H_2_O_2_, and the algal cells died after 8 days of culture [88]. The 40.95% ROS content increased in *C. vulgaris* under cetyltrimethylammonium chloride stress (0.6 g·L^−1^), resulting in structural damage to chloroplasts and decreased activity of PSII [97]. In another study, *Scenedesmus* sp. and *Chlorella* sp. produced excessive ROS in chloroplasts under the treatment of CuO nanoparticles, which oxidized the surrounding lipid and protein molecules, damaged the thylakoid membrane structure, and exacerbated the photoinhibition effect of PSII [98]. These studies have shown that excess ROS can induce lipid peroxidation and reduced lipid biosynthesis in microalgae, which may result in cell death under severe conditions. Therefore, it is very important to control the proper ROS content to promote lipid accumulation in microalgae. Related antioxidant research can help us achieve this goal. For example, in *Hordeum vulgare*, the biological characteristics and stressful resistance of *SOD* gene family have been comprehensively studied, including gene structure, chromosome localization, physical and chemical properties, phylogeny and differential expression of family members. Moreover, the changes of *SOD* gene family members in barley under different treatment conditions were further analyzed, which is conducive to improving the stress resistance of barley varieties using genetic methods [99]. However, there are few reports on SOD in microalgae, whose gene family members, physicochemical properties and role in lipid accumulation need to be explored further.

## 8. Conclusions

ROS production is an inevitable biological process in the metabolism of microalgae, especially in organelles with high electron transfer rates. The effective clearance systems can help maintain ROS homeostasis in microalgae. Various studies have shown that a certain level of ROS may activate signal transduction pathways to improve lipid accumulation in microalgae. Other studies of ROS and lipid accumulation in microalgae need to be further explored, which may provide guidance for developing techniques to enhance lipid production in microalgae based on stress strategies.

## Figures and Tables

**Figure 1 ijms-24-11041-f001:**
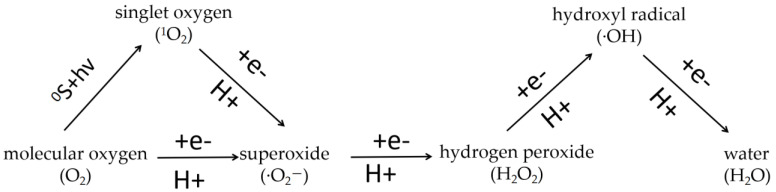
The relationships of various reactive oxygen species (ROS).

**Figure 2 ijms-24-11041-f002:**
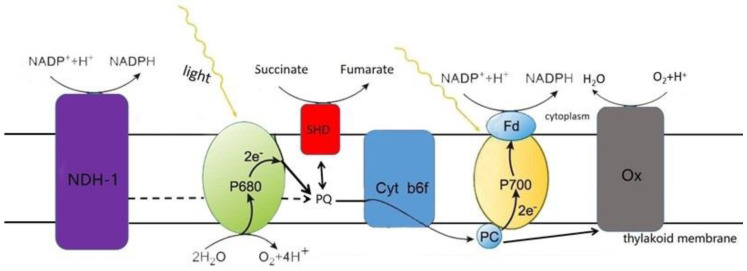
Schematic representation of the intersecting photosynthetic and respiratory electron transport pathways in the cyanobacterium *Synechocystis* sp. PCC 6803.

**Figure 3 ijms-24-11041-f003:**
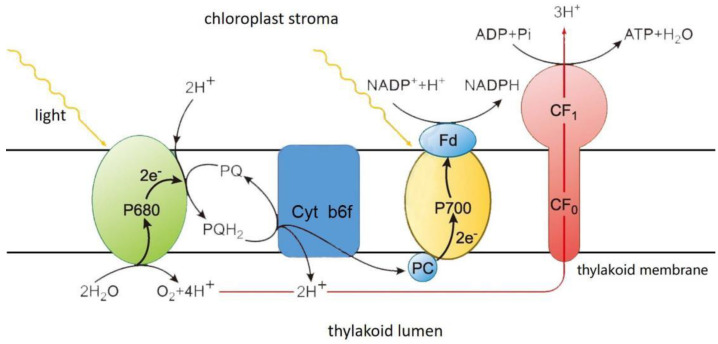
Diagram of photosynthetic electron transport pathways and the arrangement of protein complexes in the thylakoid membrane.

**Figure 4 ijms-24-11041-f004:**
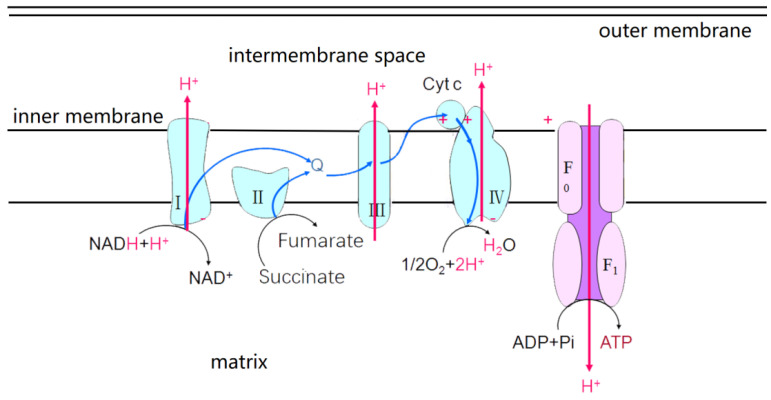
Diagram of the respiratory electron transport pathway and the arrangement of protein complexes (I–IV) in the inner mitochondrial membrane. Plus signs represent positive potentials.

**Figure 5 ijms-24-11041-f005:**
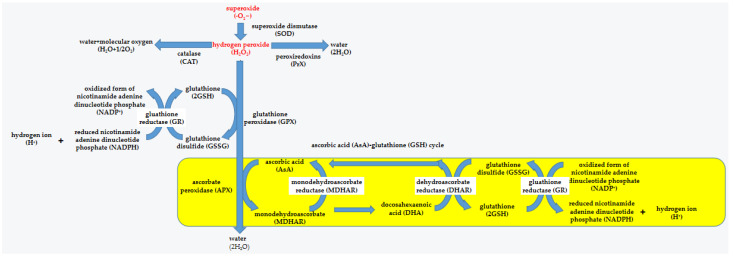
Overview of the defense system in microalgae.

**Figure 6 ijms-24-11041-f006:**
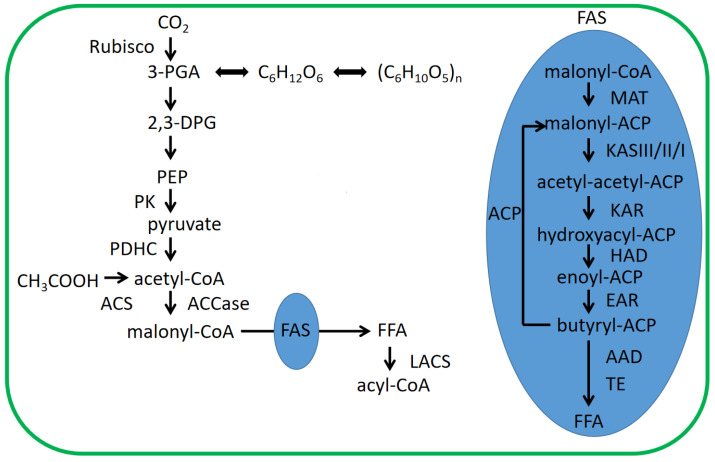
Fatty acid biosynthesis pathway in chloroplasts. 2,3-DPG: 2-phosphoglyceric acid; 3-PGA: 3-phosphoglycerate; ACS: acetyl-coenzyme A synthetase; ACCase: acetyl-CoA carboxylase; acyl-CoA: acyl-coenzyme A; acetyl-CoA: acetyl-coenzyme A; ACP: acyl carrier protein; AAD: desaturase; CH_3_COOH: acetic acid; C_6_H_12_O_6_: glucose; (C_6_H_10_O_5_)n: amylum; EAR: enoyl-ACP reductase; FAS: fatty acid synthase; FFA: free fatty acid; HAD: hydroxyacyl-ACP dehydrase; KASIII/II/I: kctoacyl synthetase; KAR: kctoacyl reductase; LACS: long chain fatty acyl-coenzyme A synthetase; malonyl-CoA: malonyl-coenzyme A; PEP: phosphoenolpyruvate; PK: pyruvate kinase; PDHC: pyruvate dehydrogenase complex; Rubisco: Rubisco ribulose-1,5-bisphosphate carboxylase; TE: thioesterase.

**Figure 7 ijms-24-11041-f007:**
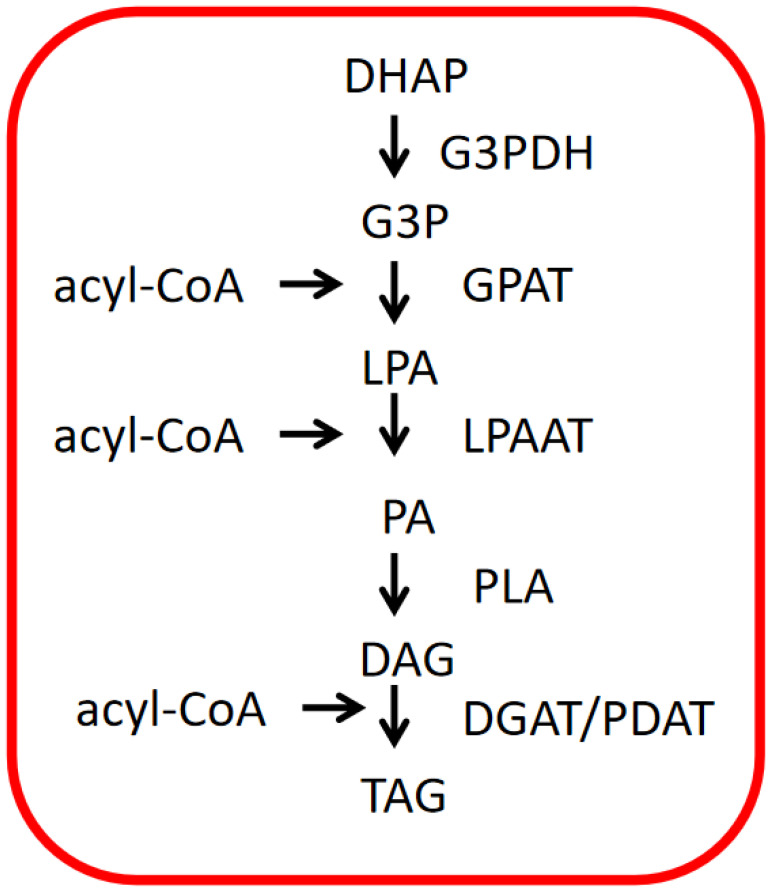
Triacylglycerol biosynthesis pathway in endoplasmic reticulum. DHAP: dihydroxyacetone phosphate; DGAT: diacylglycerol acyltransferase; DAG: diacylglycerol; G3PDH: glyceraldehyde-3-phosphate dehydrogenase; GPAT: glycerol-3-phosphate acyltransferase; G3P: glycerol-3-phosphate; LPA: lysophosphatidic acid; LPAAT: lysophosphatidic acid acyltransferase; malonyl-CoA: malonyl-coenzyme A; PA: phosphatidic acid; PLA: phosphatidic acid phosphatase; PDAT: phospholipid diacylglycerol acyltransferase; TAG: triacylglycerol.

**Figure 8 ijms-24-11041-f008:**
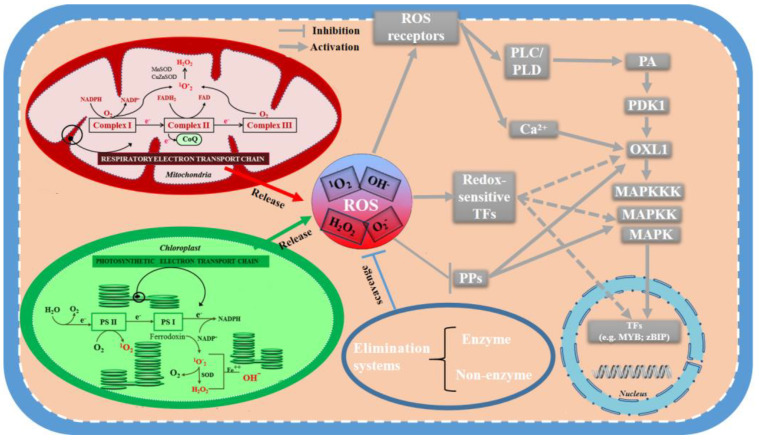
The metabolism of reactive oxygen species in lipid biosynthesis in microalgae. ROS: reactive oxygen species; ^1^O_2_: singlet oxygen; ·OH: hydroxyl radical; O_2_^−^: superoxide; H_2_O_2_: hydrogen peroxide; NADPH: reduced nicotinamide adenine dinucleotide phosphate; NADP^+^: nicotinamide adenine dinucleotide phosphate; FADH_2_: reduced flavine adenine dinucleotide; FAD: flavine adenine dinucleotide; CoQ: coenzyme Q; PS1: photosystem1; PS11: photosystem11; SOD: superoxide dismutase; PLC/PLD: phospholipase C/D; PA: phosphatidic acid; PDK1: phosphoinositide-dependent kinase1; OXI1: protein kinase OXI1; MAP(KKK): mitogen-activated protein (kinase kinase kinase); PPs: protein phosphatases; TF(s): transcription factor(s). Dotted arrow represents potential activation and coloured (i.e., red, green and blue) arrows are the effects of corresponding biochemical structures (i.e., mitochondrion, chloroplast and elimination system) on ROS.

**Table 1 ijms-24-11041-t001:** The studies of ROS-mediated lipid accumulation in microalgae.

Microalgae	Treatment	Changes of ROS Content	Changes of Lipid Content/Lipid Productivity	Ref.
*Monoraphidium* sp.	Salt stress and 1 mmol·L^−1^ H_2_O_2_	Total ROS content increased	Lipid productivity was 107.25 mg·L^−1^·d^−1^	[12]
*Acutodesmus dimorphus*	NaCl	H_2_O_2_ content increased	Lipid content increased by 43%	[13]
*Chromochloris zofingiensis*	DPI and nitrogen starvation	Total ROS content decreased	Total fatty acids and neutral lipids decreased	[14]
*Haematococcus pluvialis*	Nitrogen limitation, highlight and BHT	Total ROS content decreased	Lipid content increased by 10.71%	[15]
*Monoraphidium* sp.	Cadmium stress	Total ROS content increased	Lipid content was 52.78%	[16]
*Scenedesmus* sp.	UV and 2 mmol·L^−1^ H_2_O_2_	—	Lipid production increased by 3 times	[17]
*Scenedesmus* sp.	0, 17.64, 35.29 mmol·L^−1^ NaNO_3_ and 10 mmol·L^−1^ H_2_O_2_	—	The lipid production was 1.3, 1.2 and 1.3 times that of the group without H_2_O_2_, respectively	[18]
*Chlorella pyrenoidosa*	·OH	—	Total fatty acids and neutral lipids increased	[19]

“—” indicates that this parameter is not tested in the literature.

## Data Availability

Data sharing is not applicable to this article as no datasets were generated or analyzed during the current study.

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
