# Peer review of "The Metabolism of Reactive Oxygen Species and Their Effects on Lipid Biosynthesis of Microalgae"

_ijms, 2023, doi:10.3390/ijms241311041_

Round 1
Reviewer 1 Report (Previous Reviewer 1)
I see some modifications have been made to the manuscript and that you deleted the final paragraph of the introduction and re-written it which now reads better.
I only have one minor edit suggestion: delete 's' in "metabolisms" in title to leave 'metabolism'. Metabolism can be both singular and plural based on context, "metabolisms" just sounds wrong!
quality of English is good, no issues apart from suggestion in comments.
Author Response
Please see the attachment

Reviewer 2 Report (Previous Reviewer 2)
The manuscript is improved in some places following my recommendations. Especially the aim of the review is written more unambiguously. Also, figures are improved. Regarding figures, I recommend adding the name of particular elements of the drawings, like thylakoid membrane (Fig. 3) or matrix, inner membrane, and intermembrane space (Fig. 4).
Regarding my objections related to the spare information about lipid biosynthesis, perhaps I did not write it enough clearly, but it will be valuable to add one more chapter to the manuscript describing shortly the current stage of the art in lipid biosynthesis in microalgae, and a scheme of lipid biosynthesis would be welcomed as well. Lipid biosynthesis is one of the main two topics of the manuscript, so the article without a description of lipid biosynthesis will be defective, all the more so that the literature about lipid biosynthesis in microalgae is available. Maybe the action places of ROS will be possible for marking in the lipid biosynthesis scheme?
Round 2
Reviewer 2 Report (Previous Reviewer 2)
I appreciate the additional information on storage lipid biosynthesis in microalgae that the authors have provided in the revised manuscript. However, I have some concerns about the accuracy of Figure 6, which shows the fatty acid biosynthesis pathway in microalgae. The authors have indicated that fatty acid β-oxidation and the TCA cycle are located in the chloroplast, but this is not correct for two reasons. First, fatty acid β-oxidation and the TCA cycle are involved in fatty acid degradation, not biosynthesis. Second, fatty acid β-oxidation occurs in peroxisomes or spherosomes, and the TCA cycle occurs in mitochondria, not in the chloroplasts. The authors need to revise Figure 6 to reflect the correct subcellular localization of these cycles or provide appropriate references to support their claims.
The reference [52] must be removed from the manuscript because it is written in Chinese and this causes that it is not available for editors, reviewers and readers who do not know Chinese. Citing such references in article published in international English-language journals is not appropriate. The reference [53] is not available without a subscription, but I am afraid that it also is in Chinese.
The resolution of Figure 8 must be increased.
Author Response
Please see the attachment.

This manuscript is a resubmission of an earlier submission. The following is a list of the peer review reports and author responses from that submission.
Round 1
Reviewer 1 Report
This review is a useful summary of ROS and lipid oxidation in microalgae and should form a useful reference to those in the field. In general I though it is quite well structured apart from the last paragraph of the introduction (see my comment in attached file).

The quality of English is generally sound and only minor edits are required. I have made comments in the attached pdf.
Reviewer 2 Report
· Unfortunately, I cannot recommend acceptance of the manuscript. In my opinion, the manuscript contains too much general information about, for example, chloroplast and mitochondrion structure, electron transport chains during photosynthesis and respiration, or about ROS Generation and detoxification. The majority of the manuscript contains data that we can find in many review papers or even in academic books. It makes the manuscript can be not interesting to a broad scientific community. Table 1 and chapters 5 and 6 are interesting, but the rest of the text is too general and basal and these parts of the manuscript are not enough for a good review paper.
· Special attention was paid, for example, to chloroplast and mitochondria structure, electron and proton transport during photosynthesis and respiration, as well as to ROS production and detoxication, but nothing is written about the pathway(s) of lipid biosynthesis. Why? If one aspect of the manuscript is so detailed described (electron transport chains and ROS production and detoxification) so why the second topic (lipid biosynthesis) is so neglected?
· I am not convinced that lipid is a byproduct (Line 38).
· Each figure is prepared in a different style. It makes an impression, that there was no one main idea of the text. Additionally, Fig 6 is so simple that it is completely not necessary.
· Keywords should not be the same as words already used in the text.
· Although I am not a judge in English, the scientific language should be improved.